

# The Norwegian forecasting and warning service for rainfall- and snowmelt-induced landslides

Ingeborg K. Krøgli[1], Graziella Devoli[1,2], Hervé Colleuille[1], Monica Sund[1], Søren Boje[1], Inger Karin Engen[1]

[1]Section for forecast of flood and landslide hazards, Department of Hydrology, Norwegian Water Resources and Energy Directorate (NVE), Oslo, 0368, Norway
[2]Department of Geosciences, University of Oslo, Oslo, 0316, Norway

*Correspondence to*: Ingeborg K. Krøgli (ikl@nve.no)

**Abstract**

The Norwegian Water Resources and Energy Directorate (NVE) has run a national flood forecasting and warning service since 1989. Back in 2009, the directorate was given the responsibility of initiating also a national forecasting service for rainfall-induced landslides. Both services are part of a political effort to improve flood and landslide risk prevention. The Landslide Forecasting and Warning Service was officially launched in 2013 and is developed as a joint initiative across public agencies between NVE, the Norwegian Meteorological Institute (MET), the Norwegian Public Road Administration (NPRA) and the Norwegian Rail Administration (Bane NOR). The main goal of the service is to reduce economic and human losses caused by landslides. The service performs a national landslide hazard assessment every day describing the expected awareness level at a regional level (i.e. for a county and/or group of municipalities). The service is operative seven days a week throughout the year. Assessments and updates are published at the warning portal www.varsom.no at least twice a day, for the three coming days. The service delivers continuous updates on the current situation and future development to national and regional stakeholders and to the general public. The service is running in close cooperation with the flood forecasting service. Both services are based on the five pillars: automatic hydrological and meteorological stations, landslide and flood historical database, hydro-meteorological forecasting models, thresholds or return periods, and a trained group of forecasters. The main components of the service are herein described. A recent evaluation, conducted on the four years of operation, shows a rate of over 95% correct daily assessments. In addition positive feedbacks have been received from users through a questionnaire. The capability of the service to forecast landslides by following the hydro-meteorological conditions is illustrated by an example from autumn 2017. The case shows how the landslide service has developed into a well-functioning system providing useful information, effectively, on-time.

# 1 Introduction

Early warning systems (EWS) have been defined by UN/ISDR (2009) as "a set of capacities needed to generate and disseminate timely and meaningful warning information to enable individuals, communities and organization, threatened by a hazard to





prepare and to act appropriately and in sufficient time to reduce losses". They must comprise four elements: risk knowledge, monitoring and warning services, dissemination and communication, and response capability (UN/ISDR, 2006). A worldwide overview of existing EWS for rapid mass movements and for weather-induced landslides is available in Stähli et al. (2015) and Calvello (2017). Based on the size of the area covered by the system, landslide EWS can be separated in: a) local, that

5    focus on a single landslide at slope scale and b) territorial that focus on multiple landslides at regional scale, over a basin, municipality, region or a nation (Bazin, 2012; Calvello, 2017). Stähli et al. (2015) recognized three main categories of EWS for rapid mass movements: alarm, warning and forecasting systems (Table 1).

**Table 1.** Type of EWS for rapid mass movements and weather-induced landslides, modified from Stähli et al. (2015) and Calvello (2017)

| Type of EWS | | Explanation |
|---|---|---|
| *Local* | *Alarm* | "It detects process parameters of ongoing hazard events to initiate an alarm automatically, e.g., in the form of red flashing lights accompanied by sirens. The accuracy of the prediction is high, but the lead time is short. The alarm decision is based on a predefined threshold." |
| | *Warning* | "It aims to detect significant changes in the environment (time-dependent factors determining susceptibility with respect to mass release), e.g., crack opening, availability of loose debris material and potential triggering events (e.g., heavy rain), before the release occurs and thus allow specialists to analyse the situation and implement appropriate intervention measures. The information content of the data is often lower in this early stage, but the lead time is extended. The initial alert is based on predefined thresholds." |
| *Territorial* | *Forecasting* | "It predicts the level of danger of a rapid mass movement process, typically at the regional scale and at regular intervals. In contrast to warning systems, the data interpretation is not based on a simple threshold but is conducted on a regular basis, e.g., daily. Specialists analyse sensor data and consult models to forecast the regional danger levels, which are communicated widely in a bulletin." |

The number of existing territorial and landslide forecasting systems seems to have increased in recent years. Calvello (2017) suggests that this can be due to: better cost-effectiveness, compared to the realization of structural mitigation measures, easy applicability over large and densely populated areas where the risk to people is widespread; upgraded technologies and more reliable models in weather forecasts. However, this could also be explained by the fact that several territorial EWS working

15    operationally have started to become visible in international literature just recently, mainly in the last five years, like the EWS from Alerta-Rio, from Brazil (D'Orsi, 2012) operating since 1997. Others are still not well known outside their own region, typically due to a lack of international publication and documentation. This seems to be the case for services in Central



America, operational since the early 2000s (i.e. El Salvador, described in Devoli et al., 2015 and in Nicaragua, G. Devoli personal communication), and the Norwegian service described in this document. It is challenging for territorial and local operational EWS to reconcile typical operational tasks with research activities and dissemination of experiences to an international audience. Often, especially for territorial services; operational activities and continuous improvement of the service seems to have higher priority than publicising the latest development internationally. For some services, frequent catastrophic events may also limit the required time and attention to publish articles. Besides, documentation is often published in the original language of the service first, sufficient for the direct users, but less accessible to international readers.

The existing operative services around the world focus on prediction, warning and sending alarm to the population about possible occurrence of fast moving landslides, usually shallow, which are triggered by intense rainfall and/or snowmelt. These landslides fall in the category of flow-type landslides (Hungr et al., 2001) like debris flows, debris flood, debris avalanches, but also, translational or rotational debris and soil slides, can be observed (Hungr et al., 2014). They occur in steep slopes, usually covered by quaternary loose deposits (like tills deposits, volcanic sediments, loess, lateritic soils, etc.). Because of their long runout and high velocity, they are responsible of large damages and casualties worldwide (Dowling and Santi, 2014). In regions covered by snow, slushflows, another rapid mass movement, may also be triggered during rainfall and snowmelt episodes. Slushflows are movements of water-saturated snow which initiate in gentle slopes and are characterized by long runouts. Their high density and velocity have caused dozens of fatalities as well as the destruction of buildings and closure of roads and railways (Washburn and Goldthwait, 1958; Hestnes, 1985).

With the general name "rainfall- and snowmelt-induced landslides", herein used, we refer to debris flows, debris flood, debris avalanches, translational or rotational debris and soil slides and slushflows, because they often occur under the same rainfall and/or snowmelt episodes. They regularly occur in clusters, in large number and scattered over a large area, happening frequently together with floods.

These types of landslides cause yearly significant damages in Norway to roads and railways, buildings and other infrastructure. It is expected that climate changes, with more intense rainfall and increased temperatures, will contribute to an increase in landslide hazards (Hanssen-Bauer et al., 2017; Gariano and Guzzetti, 2016). It was estimated that every years about 200 of these events hit road sectors and about 30 hit railways (Hisdal et. al., 2017). Norway has a long tradition of building physical structures (i.e. diversion dike, tunnels, etc) to protect road and railway lines in the most critical sites. Protection measures are still useful, but their maintenance is expensive and the building operations are time consuming. The climatic and topographic conditions in Norway indicate that is an impossible task to protect 100% the national infrastructure. Therefore, forecasting and warning have become a crucial mitigation options to reduce risks.

In this document we present the Norwegian Landslide Forecasting and Warning Service (known as "Jordskredvarslingen" in Norwegian). The service, or some of its components, has been partly presented and described in conference proceedings and previous articles (i.e. Devoli et al., 2014; Boje et al., 2014a; Bell et al., 2014; Piciullo et al., 2017; Tiranti et al., submitted). The service, herein presented, can be categorized as a "territorial" EWS following Calvello (2017) and as "Forecasting and warning type" based on Stähli et al. (2015). The service is designed to predict the level of danger of rainfall- and snowmelt-





induced landslides. The service predicts multiple landslides at national scale, in particular over a region (that is commonly an administrative county or a group of municipalities) on a regular basis (every day). As for the majority of territorial systems described in Calvello (2017) also the Norwegian one herein presented is managed by a governmental institution that uses warning dissemination tools to warn multiple weather-induced hazards, including floods and snow avalanches. The service

uses specialists to analyse meteorological and hydrogeological models and forecasts, sensor data and predefined national and regional thresholds. Finally, the regional danger level is widely communicated through a bulletin.

This work summarized the efforts made in the last five years by the NVE and collaborators to design, develop, and run a nationwide landslide forecasting and warning service in close synergy with the Norwegian Flood Forecasting and Warning Service. The main purpose of this article is to describe the recent development and main components of the service, indicating

also how the service is organized and how daily assessments are performed. We present the evaluation of the accuracy of assessments and use a case study as example. Finally we present some feedbacks from regional and local emergency authorities on the usefulness of this new service.

## 2 Major floods and landslides in Norway

The mainland of Norway (Scandinavian Peninsula) covers an area of 324 000 km$^2$, with more than 490 000 km of rivers and

streams and around 250 000 lakes. The country has large climatic contrasts, from maritime to continental climate, because of rugged topography that causes large local differences. The average annual precipitation is about 1400 mm, of which about 1/3 is snow. The precipitation distribution is non-uniform. In Western Norway, annual precipitation may exceed 5000 mm and daily values of 70 mm are not uncommon. In the east, some valleys annually receive less than 300 mm. The Fennoscandian Shield constitutes the Precambrian bedrock of Scandinavia. The oldest rocks date back 2.5 billion years can be found in

Northern Norway. Above the bedrock lies remnants of the Caledonian mountain range, while the youngest rocks to be found in the Oslo Rift and provide evidence of volcanic activity 250-300 million years ago (Solli and Nordgulen, 2006). During quaternary, ice sheets covered Scandinavia several times. This resulted in poorly weathered but fractured bedrock without primary porosity, and young, sparse and thin sedimentary deposits. The aquifers in Norway mainly consist of: a) small, highly permeable glaciofluvial aquifers along streams and lakes, b) small precipitation-fed tills in mountainous areas and c) overlying

fractured bedrocks without primary porosity, such as crystalline and metamorphosed hard rocks. The tills have limited storage capacity and groundwater responds fast to water input (rain and snowmelting). There are very few large and slowly responding groundwater reservoirs in Norway. A recharge-discharge mechanism determined by the physiographic and climatic conditions controls groundwater level (Colleuille et al., 2007). In winter, precipitation fall as snow and ground may freezes. This leads to decrease of groundwater levels, increase of soil water storage capacity, and contribute to surface runoff in streams and river.

Following soil thaw and snowmelt in spring, groundwater levels rises rapidly.

Major natural hazards in Norway are extreme weather (wind storm, intense rainfalls), floods and different types of mass movements. Mass movements includes landslides, snow avalanches and slushflows. Rock fall, rock slides, rock avalanches,



mountain deformations (with a tsunamigenic potential), debris avalanches, debris flows, debris slides, rotational clay slides and quick clay slides are the most frequents landslide types in Norway (NVE, 2011). Different types of snow avalanches can be observed and slushflows are also very common rapid mass movements (Fig. 1, a-d).

The main flood types in Norway are rain flood, flood due to snowmelt, the combination of rainfall- and snowmelt-induced

flood and flash flood due to intense rainfall, the latter especially in summer. It is the combination of rainfall- and snowmelt-induced flood that historically gives the largest floods in Norway, both in return periods and extent (e.g. South-East Norway, in 1995 and 2013). In coastal areas rain flood in autumn usually give the largest floods. This especially is the case for Western Norway and the North of Norway. In some glacial valleys, jøkulhlaup (glacier lake outburst flood) is a reoccurring and potential dangerous event. Flood due to ice jams are also a phenomenon well known in Norway, both during mild periods in winter and

in springtime (Roald, 2013) (Fig. 1, e-g).

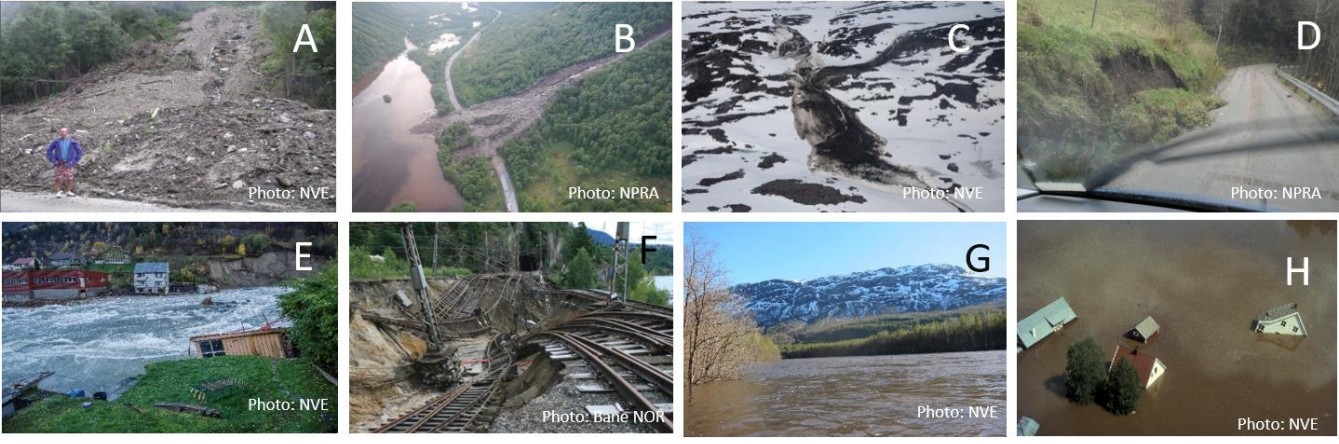

**Figure 1.** Examples of landslides and flood types in Norway. a) Debris slide. Veikledalen, Oppland. May 2011. b) Debris flow. Mjåland, Rogaland. June 2016. c) Slushflow. Troms, May 2010. d) Soil slide. Vennesla, Vest-Agder. October 2017. e) Rain flood. River Opo in Odda, Hordaland. October 2014. f) Flash flood. Notodden, Telemark. July, 2011. g) Snowmelt-induced flood. The river Reisa, Troms. May, 2013.

h) Combined rainfall- and snowmelt-induced flood. River Glomma, Hedmark. June, 1995.

Rainfall- and snowmelt-induced landslides are triggered by water. Intense or long duration water supply, caused by rain and/or snowmelt, increases the water content in the soil or snow. The cohesiveness of soil or snow particles decreases with higher water content, increasing the risk for mass transportation. Steep natural slopes covered by loos quaternary sediments, but also gentle slopes covered by snow as well as modified slopes and filling along roads and railways are especially exposed to this

kind of hazards. Climate scenarios for Norway indicate an increased occurrence of extreme weather, and intense precipitation is also expected to increase especially in the coastal areas of Norway (MET, 2013). Higher temperatures have led to earlier spring floods, and there is a tendency to increase frequency of rain floods. Future projections show that rain flood magnitude will increase, while snowmelt floods will decrease over time. More frequent and stronger intense rainfall events may in the future give special challenges in small, steep rivers and in urban areas. Weather conditions are main triggers of certain types

of landslides and snow avalanches, therefore changes in climate may thus affect their future frequency. The risk of slushflows will increase, and may occur in areas where they have not occurred previously (Hanssen-Bauer et al., 2017).





The experience acquired from landslide events in Norway since 2011 shows that they can occur all year around. Numerous events have happened often in association with floods events, producing substantial damages to roads, railway lines, buildings, and other infrastructures. Important recent past landslides events occurred in November 2000, August 2003, September 2005, November 2005, Mai 2008, June and December 2011, March 2012, July 2012, May 2013 and Sept/Oct 2017.

There are limited comprehensive estimates of human and economic losses associated to natural hazards in Norway (e.g. floods and mass movements). In terms of fatalities, about 2000 persons have lost their live in the past 150 years because of mass movements. Most of these casualties are due to snow avalanches (Nadim et al., 2008). Within the period 2009-2016 it is registered 54 fatalities from snow avalanches alone (http://www.varsom.no/ulykker/snoskredulykker-og-hendelser/; http://www.varsom.no/ulykker/). For landslides in soil, Aaheim et al. (2010) reported that 100 people died since 1900 and

most of the casualties are related to clay slides and quick clays slides, often triggered by anthropic factors. Few data are available for casualties related to rainfall- and snowmelt-induced landslides. An effort to document fatalities associated to these landslide types was done by NVE in 2016 as part of the work presented in Haque et al. (2016) where landslides fatalities have been presented for the entire Europe. For Norway the analysis showed that 42 people died in the period 1995-2016 due to 25 landslide events in the category of debris flows, debris avalanches, clay slides, quick clay slides, rock falls, rock

avalanches and slushflows. The results indicated that 2005 and 2010 were the years with most recorded fatalities (ca. six persons). Most of the fatalities were caused by rock falls and rock avalanches, seven because of clay types slides, while 12 people died because of rainfall- and snowmelt-induced landslides (of these seven due to slushflows and five due both debris flows and debris avalanches).

In terms of economic losses, there are no reliable estimates of the total cost to society due to natural hazards, although insurance

payments can provide an indication of cost trends. Payments made by insurance companies in Norway between 1980 and 2014 show both an increase number of damaging events and increase number of total claims per year, reaching around 2 500 mill NOK in the flood and landslides event of June 2011 in south-eastern Norway. However these number are underestimated since they do not include events and costs associated with public infrastructure (NIFS, 2016).

**3 The Norwegian landslide forecasting and warning service**

**3.1 History, mandate and organization**

NVE is a directorate under the Ministry of Petroleum and Energy and is responsible for the administration of Norway's water and energy resources, and the coordination of national efforts for landslide and snow avalanches risk prevention. NVE operates three forecasting services (landslide, flood and snow avalanche) and several local warning system for large rockslides (Engeset,

2013; NVE, 2017; Blikra and Kristensen, 2013). The three forecasting services are operated by the Hydrology Department, responsible for collecting, storing and analysing hydrological data. The Landslide, Flood and River Management Department operate the monitoring and warning system for large rockslides.





Already in 1804, after the great flood in 1789, the first national river management agency of Norway was founded, and was the precursor to NVE. The first spring flood warning was implemented already in the late 1960s for the Glomma River, the largest river in Norway, however, it was only in 1989 that the flood forecasting service started to be operational for the entire country. In 1995 a major flood resulted in huge damages and this contributed to increase resources and commitment to the

development of new technologies to improve the service.

In 2009 NVE was assigned to support regions and municipalities in the prevention of disasters posed by landslides and snow avalanches in addition to flood, as a part of governmental effort to improve public safety (White papers: Meld. St.22 (2007-2008); Meld. St. 15 (2011-2012)). Main tasks and activities include the collection of landslide data and implementation of digital mass movement inventories; preparation of susceptibility and hazard maps; assistance in areal planning; elaboration of

regulations and guidelines; financial assistance in the construction of protection measures; organization of early warning systems and assistance during flood, landslides and snow avalanches emergencies.

The development of EWS for landslides and snow avalanches started in February 2010 based on suggestions in Colleuille and Engen (2009). The snow avalanche forecast service was launched in January 2013 and follows European standards as it is part of the European Avalanche Warning Service (Engeset, 2013). A new organisation at the Hydrology Department at NVE in

2011 lead to higher priority for the flood forecasting and the development of the landslide forecasting service. The landslide forecasting service started an operational test phase in January 2012. This service was officially launched in October 2013 and is running in close cooperation with the National Flood Forecasting and Warning Service.  Since then, the service has operated continuously at regional scale for mainland Norway. The service is developed as a joint initiative across public agencies between NVE, the Norwegian Meteorological Institute (MET), the Norwegian Public Road Administration (NPRA) and the

Norwegian Rail Administration (Bane NOR).

**3.2 Components**

A sustainable EWS for rainfall-induced landslides requires: strong and reliable meteorological, hydrological, hydrogeological, or geotechnical models; meteorological, hydrological, hydrogeological and geotechnical networks; a national landslide database to support threshold development, probability analysis, and verification; geographically specific warning thresholds;

a uniform, national scale shallow susceptibility map or hazard map; computer and communications networks to support the operation and an operational infrastructure and dedicated professional staff. Political commitments and dedicated investments are also crucial. The service need to be integrated part of national and local disaster risk management plans and budgets and enforceable legislation must defines roles and responsibilities of local to national authorities and agencies involved. Because of the multidisciplinary characteristic of these types of landslides the cooperation among agencies should be effective. Finally,

the service requires systematic feedback and evaluation at all levels to ensure improvement, implementation/commitment over time and systematic field verifications (UN/ISDR, 2006). The main components of the Norwegian landslide forecasting and warning service are described in the following chapters.



### 3.2.1 Meteorological forecasts and hydrological models

The service use daily meteorological quantitative gridded forecasts of precipitation and temperature, obtained from MET. The forecasts are obtained from different weather models: AROME-MetCoOp (short-term forecasts used in the Scandinavian regions in cooperation MET-Norway with Swedish Meteorological and Hydrological Institute and Finnish Meteorological

Institute) and EC, which is a global long term model from the European Center for Medium-Range Weather Forecasts (ECMWF). The short term model's resolution is 2.5 km, and is used for the + 66 h forecast and is updated four times a day. The long term model's resolution is 9 km, forecasts for nine days ahead and is updated twice a day.

Due to the sparse station network and relative short measurements periods, hydrological models are a requisite to describe the water and energy balances on a national scale. The service uses forecasted hydro-meteorological variables obtained by a

distributed version of the hydrological HBV-model (Beldring et al., 2003). The model divides Norway into 1 km$^2$ grid cells (total over 385 000 cells), where each cell is treated as a separate basin with a corresponding simulation of the water balance. The model simulates for example runoff, snowmelt, groundwater, soil saturation and soil frost, based on two input data, temperature and precipitation. Forecasted values are obtained from downscaling of the AROME and EC weather prediction models, while observed values are based on interpolated  values from MET's nationwide network. The model is automatically

running four times per day. Several of the models simulated variables can be found at www.xgeo.no as maps (see chapter 3.2.6).

We use, in addition to the distributed HBV-model, a one dimensional soil water and heat flow model (S-Flow) developed by NVE. This model simulates water and heat dynamics in a layered soil column covered by vegetation. S-Flow used equations adapted mostly from the COUP (Jansson and Karlberg, 2014) and SHAW (Flerchinger, 2000) models. The model run with a

daily time step, using precipitation, air temperature, wind speed, relative humidity and sun radiation (or cloud cover) data as input. In addition, plant growth characteristics and soil characteristics are necessary inputs to the model. Simulations with the S-Flow are performed only in areas where groundwater stations are located (about 45 points), where observations are used for parameterisation of the model. The model runs daily and the results, as water supply (snowmelt and rain), soil water deficit, groundwater level and soil frost, are available at xgeo.no (see chapter 3.2.6). S-Flow model has a better physical description

than HBV model of the snowmelt and evaporation process as it uses a physically based approach and all available meteorological information. In addition to the estimation of soil-water deficit, S-Flow includes soil-water depletion following the fall in groundwater levels in winter caused by lack of recharge and groundwater discharge into streams and lakes (Colleuille et al., 2007).

### 3.2.2 Meteorological and hydrological network

The service uses several networks. We can access data from meteorological stations, equipped with rain gauge (hourly and daily data), temperature sensors and snow and wind sensors, and operated mostly by MET, but also by NPRA and Bane NOR.





Hydrological stations are used to measure discharge in rivers, snow depth and coverage (over 400 stations) and hydrogeological stations to measure groundwater level (70 stations) and are operated by NVE.

Real time observations of rainfall, air temperature, water discharge and ground water level are used in the daily landslide hazard assessment to check the performance of the hydro-meteorological conditions obtained from the hydrological models.

This is particularly important when the models overestimate or underestimate certain parameters values (i.e., the soil water saturation or the snowmelt) in certain regions or in certain seasons. Real time discharge data are used to automatically assimilate and correct the modelled discharge in watercourses and are most useful for flood forecasting, but can be also give valuable information for the debris flows hazards. Historical data on soil moisture, soil frost and groundwater have been mainly used to test and calibrate the physically based model S-Flow.

**3.2.3 Landslide database**

Landslide records are essential for different types of analyses, e.g. threshold establishment, calibration of models in warning systems and evaluation of warning performance. Landslide data can be collected using two interfaces regobs.no (see chapter 3.2.9) and www.skredregistrering.no. This last one is the web portal for the national mass movement database, containing landslides and snow avalanches events. The database contains more than 50 000 events in the categories of rock fall, rock

avalanche (of different sizes), debris flow, debris slide or shallow soil slide in artificial slopes, snow avalanche, icefall and landslide in clay (quick clay slides and rotational clay slides). In addition some events can be recorded as unspecified when the subtype is unknown. The database is maintained by NVE, but many institutions can registered data among them the NPRA, Bane NOR, the Geological Survey of Norway (NGU) and the Norwegian Geotechnical Institute (NGI). The data are accessible through NVE Atlas and xgeo.no (see chapter 3.2.6). The landslides are represented by points positioned where the event caused

losses of life, damages or traffic interruptions. The database contains valuable information for thresholds analyses, however, because of the many limitations, a quality control is always performed before any type of analysis.

**3.2.4 Thresholds**

The forecasting service is based on proven relationships between the time of past landslide events and meteorological and hydrological variables. The development of the forecasting system is based on the principle that since hydro-meteorological

parameters can be predicted, forecasting of landslide hazard is possible. The knowledge of these relationships is used to develop threshold values. Modelled hydro-meteorological variables obtained with the HBV model and cross-checked with the time of previous landslides were used to statistically derive thresholds for different regions of the country (Colleuille et al., 2010, Cepeda et al., 2012; Cepeda 2013a; 2013b; Boje et al., 2014b; Boje, 2017). The best performance was obtained when the simulated water supply (e.g. rain and snowmelt) and the simulated soil water saturation degree were combined. The

simulated degree of soil saturation (%) describes the relationship between todays total soil water content (groundwater and soil water) compared to the maximum soil water content simulated in the reference period 1981-2010. The simulated water supply is presented as percent of yearly normal water supply in the period 1981-2010, and is the product of either simulated





rain, in case there is no snowpack, or water drainage from the snowpack as a response to rain and snowmelt percolating through the snow. The threshold is called Hydmet (e.g. hydro-meteorological index). Beside the thresholds for the entire country we have developed regional thresholds (Boje et al., 2017) and all thresholds are visualized in form of raster data (with 1 km$^2$ resolution) and available at xgeo.no (see chapter 3.2.6).

### 3.2.5 Susceptibility maps

A susceptibility map shows the spatial probability of landslides, e.g. the probability that a region will be affected by landslides given a set of terrain conditions. We have sponsored the elaboration of two maps that can be used to predict the spatial occurrence of rainfall- and snowmelt-induced landslides in Norway, both of them cover the entire country. The first map,

realized in collaboration with NGI shows which 1$^{st}$ order catchments are more susceptible to landslide in soil (e.g. debris avalanches, debris flows, shallow soil slides, clay slides and quick clay slides). The map was prepared combining different variables, like quaternary cover map, land cover, average yearly rainfall, various water runoff variables, and various derivatives from the 15 m x 15 m digital elevation model (DEM), i.e. slope and aspect. It was done using the Generalized Additive Models (GAM) (Fig. 2a). This map has been used to improve the original threshold map (see chapter 3.2.4), by including information

on landslide prone-areas and the result of this combination was a new threshold map, called "Hydmet Geo". This is used by the forecasters in the initial phase to perform a more accurate assessment (Bell et al., 2014).

The second susceptibility map, realized by NGU, shows specifically where debris avalanches and small debris flows may occur at 1:50 000 scale (Fischer et al., 2012; 2014). The map displays the modelled potential source areas, tracks and runout areas. The source areas were discriminated based on an index approach, which includes topographic parameters, obtained from

20 10m DTM (i.e. slope angle, planar curvature) and hydrological settings (i.e. drainage area). For the runout modelling, the Flow-R model was used, which is based on combined probabilistic and energetic algorithms for the assessment of the spreading of the flow and maximum runout distances. This map is used in the communication phase of the warning, since it can be visualized in varsom.no (see chapter 3.2.8) together with the warning zone and warning level. The user can zoom in the map of the warning zone, and see where landslide could occur (Fig. 2a).



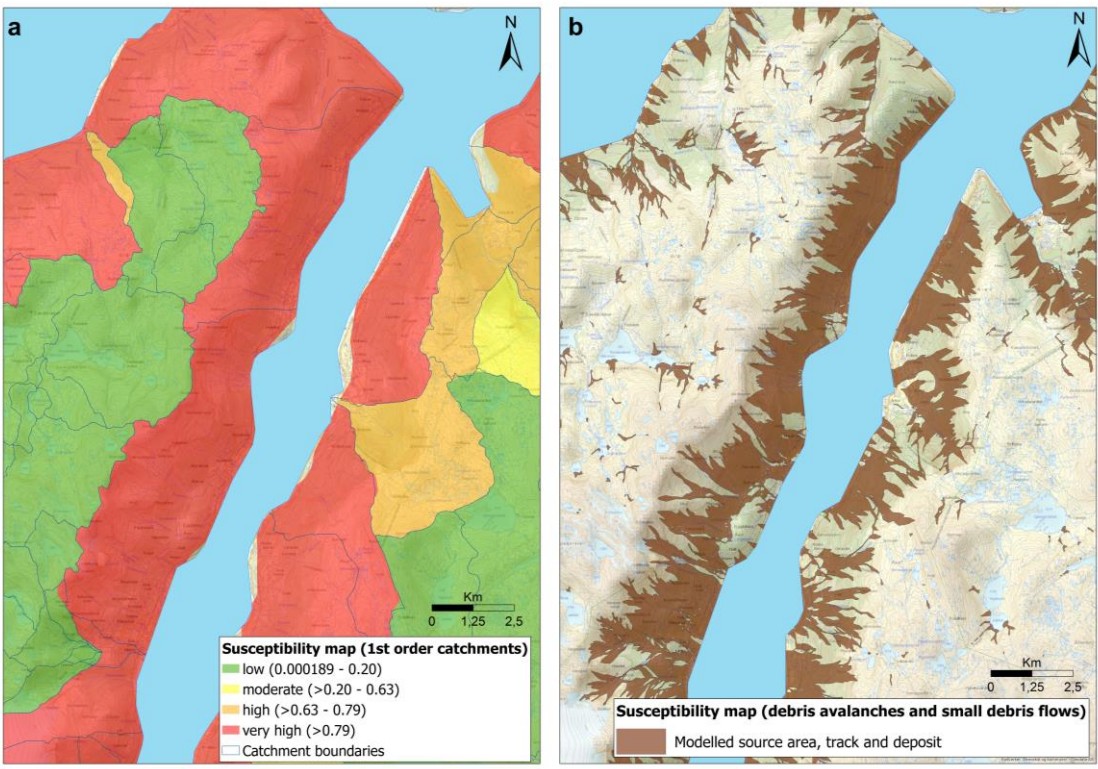

**Figure 2.** Susceptibility map for landslides in soil at Sørfjorden, Western Norway: a) at 1st order catchments from Bell et al., (2014); b) at 1:50 000 scale, from Fischer et al. (2014).

### 3.2.6 Web tools: xgeo – an analysis and decision making tool

Xgeo.no is a decision making tool used by snow avalanches, landslides and flood forecasters. Even though its use is reserved for the specialists, data is made available to the public thanks to open data policy through www.senorge.no (Engeset et al. 2004). The web portal, developed and maintained since 2008, is a map centric tool for visualization of temporal and spatial data (Barfod et al. 2013). The portal shows daily observations and forecasts for meteorological and hydrological conditions as thematic maps and time-series data. The maps, updated four times a day, show the conditions for each day, and for six days

ahead and reach back to 1957. Landslide specialists use this tool during the daily evaluation to visualize e.g. real time measurements, weather forecast, threshold values predictions, water supply and groundwater simulations, data from the real-time database regObs (see chapter 3.2.9), landslide events from the national mass movement database (see chapter 3.2.3), roads closed because of landslides and other administrative data, such as existing infrastructure (Devoli et al., 2014).

The hydro-informatics team at NVE has developed xgeo.no, in cooperation with the MET, NPRA, Bane NOR and the

Norwegian Mapping Authority (Kartverket). The tool xgeo.no is in systematically updated.



### 3.2.7 Operational infrastructure and staff

The organisation of the landslide forecast service rests heavily on the organisation of the flood forecast service. It was important to maintain and not disorganize the well-functioning flood forecast service during the development of the landslide service. This was ensured by establishing a parallel group of landslide forecasters. The landslide forecasting team consists of people

with different backgrounds, such as hydrologists, geologists, geophysicists, hydrogeologists and physical geographers. The team consists, in 2017, of twelve employees from NVE and two from NPRA. Five of the landslide forecasters work also as flood forecaster and two of them as snow avalanche forecasters. Landslide and flood forecasters discuss closely the daily landslide and flood assessments. This synergy effect, lead to improvements and strengthening for both services. The assessment of slushflows is done in collaboration with the snow avalanche forecasting service, which provides additional

information on snow structure and snow condition. There is still an ongoing effort to synchronize the three services groups where possible, and ideas and information are exchanged.

The service is operative seven days a week, throughout the year, with a rotating scheme with one forecaster on duty. The forecasters are resting at home outside the normal work time, but they can be always reached by mobile phone (8-21). Forecasters may have to be available 24/7 when there is a severe situation. In "standby situation", after the analysis of the

situation and warning updating, the forecaster on duty may work with research activities or other tasks. Forecaster, when on duty, have the possibility to work from home during weekends. Courses and training workshops are yearly organized to educate landslide forecasters, discussing new tools and exchange ideas. Many of these courses and workshops are organized together with flood forecasters as well.

Beside available and dedicated personnel as forecasters, the service benefits from skilled IT-personnel, also with strong

dedication. The real time network and forecasting tools are set up with redundant systems. In the case of failure of internet, routines have been developed to secure minimum communication both to ensure meteorological data and to convey the resulting possible heightened warning level and situation report to the public.

### 3.2.8 Communication network: varsom.no, SMS and CAP

www.varsom.no is the national web-portal for flood, landslides, snow avalanches warning and ice conditions on regional scale.

The web was chosen as the main channel for communicating bulletins and warning levels to end-users according to the decision on open access. During the development, was given high priority to the accessibility on mobile screen, according to the need of making bulletins available to the users "on site" and because of the rapidly increasing numbers of smartphone users (Johnsen, 2013). The web portal both displays bulletins and related maps for the natural hazards covered by the NVE's forecasting and warning services, but also provides additional information on precautions, educational literature and videos

and relevant reports. Through varsom.no the landslide service delivers continuous updates on the current situation and development to national and regional stakeholders and the public. Assessments and updates are published at least twice day





and contain the forecast for the three coming days. The landslide forecast is valid from 7AM the day of publication to 7AM the following day (8AM to 8AM Daylight Saving Time).

Internally the software regVars has been developed to enable the publication of flood and landslide bulletins in www.varsom.no. It provides possibilities of drafting bulletins before they are published, enabling ample time for preparation

and quality assurance. The bulletins for all three forecasting services is available on api.nve.no free of charge. From early 2017, it has been possible to subscribe to warning messages published on www.varsom.no. The subscription available at https://abonner.varsom.no is easily managed and free of charge. The users chose for themselves which natural hazards they want to subscribe for (e.g. flood, snow avalanche or landslides) and on what level they want to receive a SMS or e-mail (or both). It is possible to subscribe for all of Norway, for landslide, flood and snow avalanche, for all warning levels, or just for

one municipality and one hazard. All local and regional emergency authorities are encouraged to subscribe. In the case of orange and red level, NVE uses a crisis information management tool, called CIM, to notify by e-mail to the actual county's emergency division that warnings have been sent. The county has the responsibility to forward the message to the respective municipalities. MET, NPRAs traffic service, and NVEs regional offices are also contacted via CIM.

In 2017 NVE and MET started a project in order to use the Common Alerting Protocol (CAP) for distributing warning

notifications on severe/extreme weather, floods, landslides and snow avalanches and to try to harmonise warning procedures and products. The use of CAP is the first at its kind in Norway, and serves as the start of a Norwegian standard (CAP-NO) which may be used other types of alerts. The goal of this project is to improve communication and effectiveness of the warning services.

### 3.2.9 Verification of landslide occurrence: regObs (a crowdsourcing tool) and media monitoring

The success of landslide forecasting depends on the registration of landslide events. Landslide events are used for both the development of thresholds and the evaluation of a sent warning to confirm if the warning was correct or not. Therefore is extremely important to confirm that a landslide event has occurred after a specific triggering rainfall event (Devoli, 2017). We use different sources to verify the occurrence of landslides. regObs.no (the abbreviation for "register observations") is a real time registration tool for observations, danger signs and events to be used by forecasters and emergency personnel (Ekker et

al. 2013). In the start-up of regObs in 2010, the database was a tool for the submitting and sharing of snow avalanche observations. Later, the real time database was extended to register observations related to other natural hazards like landslides, floods and ice conditions. It was designed as a public tool supporting crowd sourcing, which means that everyone may contribute with observations and all data are immediately available to the public on the regObs website and as an app (www.regobs.no). Both NVE and NPRA stand behind the development of regObs. The data are treated as initial information,

and are subsequently checked and quality assured before they are stored in the national mass movement database (see chapter 3.2.3) and flood database.

Information from local or national newspapers provide one of the fastest sources for obtaining data on landslides affecting infrastructure. Tools for media monitoring of events is, therefore, also used as a part of the daily routine to evaluate the issued



warning levels. Furthermore, media are followed by the forecaster on duty to collect as much as landslide information as possible. Nevertheless, the accuracy on the reported event may be poor, therefore a detailed aftermath examination of the facts is essential. One source of error is the large variety in media coverage of such events, which is closely related to the hazard impact to infrastructure and population of the area. Events in more sparsely populated areas may not be covered by this

information source. Besides media, we can collect landslide information through landslide specialists working at NVE's regional offices and landslides specialists from NPRA and Bane NOR that monitor and report landslide events, after field surveys.

## 4 Daily assessment and warning levels

The daily landslide hazard assessment is performed by a forecaster who uses forecasted thresholds, forecasted hydro-

meteorological parameters, information from real-time observations, knowledge on historical events and regional susceptibility and personal experience. The daily routine is summarized in Fig. 3, and include the following phases:

- Weather forecast, also as input for the hydrological model

- Model run, forecasted hydro-meteorological parameters, forecasted thresholds

- Collection of real-time data

- Interpretation of model results. Use of additional information from simulated hydro-meteorological parameters i.e. snow and groundwater conditions

- Analysis of forecasted thresholds also corrected with susceptibility information

- Preparation of forecast information and warning messages with description of what may happen and expected impact

- Communication and dissemination of messages to warn the public and local authorities

- Provide hydrological situation updates and answer questions from media or another recipients





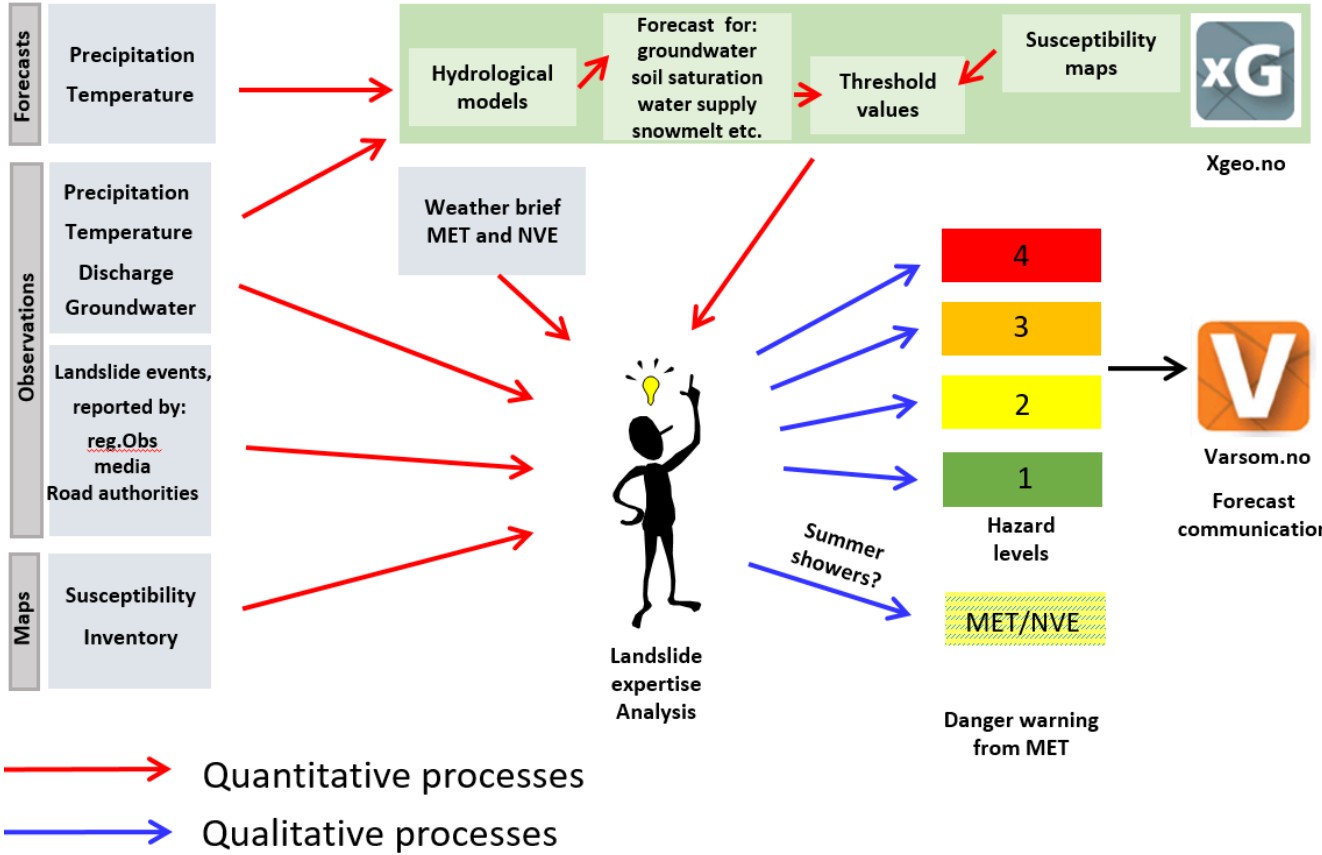

**Figure 3.** Synthesis of the how the daily landslide hazard assessment is performed (modified after Piciullo et al., 2017).

The warning scale is applicable for both flood and landslide hazards and consist of four levels using the same concept of

meteoalarm (www.meteoalarm.eu). The different levels show the level of landslide hazard and the recommended awareness

levels (Table 2), providing information on what is expected to occur, the severity (qualitative estimation of numbers and

dimensions of landslides) and recommended actions that the users should undertake or which measures should be initiated in

order to reduce potential damages (Fig. 4).

The principle behind the awareness levels is that the highest level (red) occurs very rarely while the second lowest level

(yellow) occurs more often. Just for comparison, the red level corresponds to a flood with more than 50 years of return period

while the yellow level to a flood with 2-5 years of return period.

Emergency response authorities should be prepared to implement emergency plans, considering available resources,

implementing preventive measures, safeguarding exposed assets, carry out evacuations and other contingency responses. One

of the mitigation measure recommended is to ensure unhindered water channels, e.g. that culverts are not obstructed by ice,

snow, sediments or other matter.



**Table 2.** Awareness levels used in the Norwegian landslide forecast and warning service

| Significance of the awareness levels | |
|---|---|
| **Red awareness level** | Very high landslide hazard. Many landslides and several large ones may occur; their long runout and extent may result in damage to settlements and infrastructure. Red awareness level is an extreme situation that occurs very rarely. Safety measures such as closed roads and evacuations can occur on short notice. Emergency response authorities should have implemented emergency plans, mitigation measures, carry out evacuations and other contingency responses. Pay attention to the media and follow recommendations from the authorities. |
| **Orange awareness level** | High landslide hazard. Many landslides and some large ones that can damage infrastructure and roads may occur. Exposed roads may be closed off. Emergency response authorities should be prepared to implement emergency plans, mitigation measures, and evaluate the needs for evacuations and other contingency responses. Mitigation measures such as clearing water channels should be carried out. Pay attention and follow recommendations from the authorities. |
| **Yellow awareness level** | Moderate landslide hazard, primarily shallow slides on artificial slopes that may affect roads, railways or along river embankments. Isolated debris avalanches or debris flows can occur, and could cause damages to infrastructure and people. In this level emergency authorities should increase vigilance related to landslides and pay attention to weather forecasts and landslide forecasts and information on www.varsom.no. Preventive measures are recommended, such as clearing water channels in exposed areas. |
| **Green awareness level** | Generally safe conditions. Debris avalanches, debris flows, shallow slides and slushflows are not expected at this level, however other landslide types (like rock falls, clay slides and quick clay slides) may occur, caused by slow response processes, such as erosion, freeze-thaw weathering or human activity, such as deposition, digging or blasting. These incidents may occur at all awareness levels. |





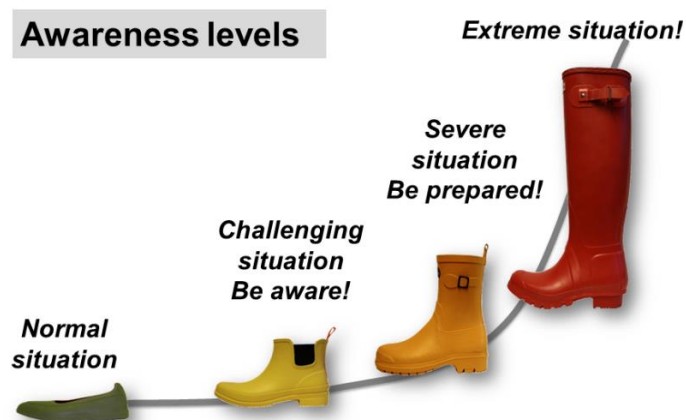

**Figure 4.** Popular representation of the awareness levels, symbolised by boots.

## 5 Validation of the forecasting service

Golnaraghi (2005) and UN/ISDR (2006) expressed that "one of the most effective measures for disaster preparedness is a well-functioning early warning system that delivers accurate information dependably and on-time". Therefore a useful EWS is the one capable to issue correct warning messages, being understood and early enough to lead municipalities and contingency planners, emergency authorities to action in order to avoid or reduce damages due to landslides. This imply that a successful service requires periodically assessments of the technical performance and the user perception (Devoli, 2017). In our service,

we evaluate the so-called technical performance and the user perception.

**Technical performance**

The technical performance is evaluated by measuring the accuracy of the service, i.e. quantifying how well the landslide warning performs (correct alarms, false alarms, missed events, wrong levels). It is assumed that a good-time service will be

perceived as credible and will trigger activity/action by users. Bad hits, with many false alarms and/or more unannounced events, will have the opposite effect. The performance of the forecast rely on the occurrence of landslides during that particular triggering rainfall/snowmelt event. The evaluation of the performance is done using a method still under development. The evaluation of the performance is done in light of the fact that the landslide warning is a regional service that warn for landslides over a larger area. A weekly evaluation of the hazard assessment is carried out in light of observed situation, number of

landslides events and confirmed hazard level. The classification criteria were presented in Piciullo et al. (2017). The evaluation is quality assured several weeks later using updated information about how the situation was, against what actually happened and registered events. Evaluation is challenging because it must be based on subjective qualitative assessments. Unclear recordings of events due to missing reporting, incorrect reporting type, date or place, make the evaluation difficult. The





performance evaluation described here reflects four factors: How good are the threshold values? How good are the hydrological simulations? How good are the weather forecasts? How well did the forecaster in duty assess the situation? The results of the preliminary analysis conducted at the national scale for the period 2013-2016 and using all days in the years, shows that over 95% of the days assessment are considered as correct. The performance is on the same magnitude for the flood forecasting

service. False alarms and unexpected events are due in most cases to changes in weather forecasts. Some false alarms and unexpected events are also due to errors in the hydrological models or incorrect interpretation of the model results.

Besides this method, the performance of the landslide service was also tested with the method Event, Duration Matrix, Performance (EduMaP) proposed by Calvello and Piciullo (2016). The method has been adapted to the Norwegian landslide forecasting and warning service (taking into account the variable warning areas) and tested for Western Norway for the years

2013-2014. Results are presented in Piciullo et al. (2017). The results show an overall good performance of the system for the area analysed.

Based on the results from both methods we have started to work on the regional improvement of landslide threshold, like it was done for the Southern and Eastern Norway, contributing in the reduction of false alarms in these regions (Boje, 2017).

**User perception**

A warning, if correctly received and understood, should contribute to a better preparedness and generate a series of actions. User surveys will provides the basis for an assessment of the value of the service. How do we best use the forecasts, and other products, prepared by the service? How do we communicate the risk?

We have performed two evaluations among users. The first survey was conducted among emergency response officers in the

municipalities, county deputy chiefs and infrastructure owners, such as the NPRA and Bane NOR for a sample of 588 people (Epinion AS, 2017). We ask among other, "How important for the user is the NVE landslide forecasting?" and "How much the user trust the NVE landslide forecasting?" Results shows that a large majority of users consider the landslide forecasting service useful or very useful and they have quite or very much confidence with the warning notifications published at varsom.no.

The second one was conducted among a working group, with personnel from NVE, MET, NPRA, Bane NOR and a County Emergency Office, that was assigned to conducted an evaluation of the snow avalanche and landslide forecasting service (Hisdal et al., 2017). The group made the evaluation based on the following criteria: development of the services, how the services work today, costs, benefits for the users, measures to improve the benefits, analysis of number of snow avalanches and the synergy between flood and landslides services. The working group concluded that the landslide service contributes to

better secure society. To improve the accuracy of the notifications, and utility of the service, four priority areas are recommended: increase communication and build capacity among users, improved hazard assessment, improve models and tools and better landslide occurrence verification.




# 6 Case study: Southern Norway, autumn 2017

Southern Norway is the area that include the counties of Rogaland and Agder (e.g. Vest-Agder and Aust-Agder). As indicated in Devoli and Dahl (2014) and later in Devoli et al. (2017) the region is characterized by predominant hills and low relief with gentle slopes (<25° locally up to 45°) along the coastlines, and moderate slopes to elevated hills (<25°, but locally 25-45°) in the interior. Alpine relief and steep slopes are observed in the valleys oriented N-S direction. The area is covered with tills, but along the coast and in the alluvial plains, the soil coverage is thicker, with fluvial deposits, used for agriculture. In the eastern parts of Agder there is also marine deposits near to the coastline.

For landslide forecasters this areas has been a challenging one since the start of the operations. The region is known to be an area with low landslide records, even though it may receive much rain in autumn and sometimes in summer and in winter. Along the coastline few debris slides and some soil slides in artificial slopes, have been registered in the database (www.skredregistrering.no). Many of these records are often lacking of detailed information (i.e. unknown landslide type, day of occurrence, etc.), and many were not triggered by natural causes (i.e. rainfall/snowmelt) and occurred in days without rainfall, possibly triggered by anthropic causes. In the interior of the region, records of debris flows and debris slides are almost absent and, the few ones have poor quality and are very uncertain. The lack of landslide records is due also to the low density population and of transportation lines. From the experience acquired in the last 5 years and evidenced as well by the warning performance evaluations realized so far (see chapter 5) it was clear that the thresholds were too high for the area. In 2016-2017 we reviewed the thresholds and tuned and updated them based on a few recent but most reliable events (Boje, 2017).

At the end of September and beginning of October 2017 two powerful low pressure systems, located initially north of Newfoundland, brought intense rainfalls over 3-4 days starting on the 29th of September 2017. The first low pressure system was supposed to hit the western sector of the region, while the second one that also carried the rest of the tropical cyclones Maria and Lee was supposed to hit the eastern part of the region, including also the Telemark county and even some of the counties in south-eastern sector of Norway. MET had warned NVE on these weather systems with some days in advance. MET released a meteorological warning for the region based on available forecasts. A flood and a landslide warning were also issued (Table 3).

The flood forecasting issued a warning at yellow level for Saturday 30th September on Thursday 28th September. The flood warning was levelled up to orange level for Saturday 30th September on Friday 29th. On Saturday 30th the flood warning level was set to red, and stayed on red level for most of the Agder counties for three days, followed by an orange day (3th Oct.) and one day yellow (4th Oct.) before the rivers were down to normal levels (Table 3).



**Table 3.** Daily assessments and issued flood and landslide warnings for Southern Norway, between 30/9 and 4/10, 2017.

| Landslide: Day when the assessment was performed | 30th | 1st | 2nd | 3rd | 4th |
|---|---|---|---|---|---|
| 3rd | | | | Rogaland and Hordaland | |
| 2nd | | | Agder and Rogaland | Agder, Telemark and Rogaland | |
| 1st | | Agder and Telemark and southern part of Rogaland / Southern parts of Agder | Agder, Telemark and Rogaland | Agder, Telemark and Rogaland | |
| 30th | Agder and Telemark / Southern parts of Agder | Agder and Telemark | | | |
| 29th | Agder and Telemark | Agder and Telemark | | | |
| 28th | E-mail to NPRA and Bane NOR with information on the coming weather | | | | |
| | **30th** | **1st** | **2nd** | **3rd** | **4th** |
| | | | **Day for which the warning was valid for** | | |

| Flood: Day when the assessment was performed | 30th | 1st | 2nd | 3rd | 4th | 30th | 1st |
|---|---|---|---|---|---|---|---|
| 3rd | | | | | | Southern parts of Rogaland, north of Agder / Southern parts of Agder | Southern parts of Rogaland and Agder |
| 2nd | | | | Telemark / Southern parts of Rogaland, north of Agder and Telemark / Southern parts of Agder | | Southern parts of Rogaland, north of Agder / Southern parts of Agder | Southern parts of Rogaland and Agder |
| 1st | | Telemark / Southern parts of Rogaland, north of Agder and Telemark / Southern parts of Agder | | Telemark / Southern parts of Rogaland, north of Agder and Telemark / Southern parts of Agder | | Southern parts of Rogaland, north of Agder and Telemark | |
| 30th | Southern parts of Rogaland, north of Agder and Telemark / Southern parts of Agder | Southern parts of Rogaland, north of Agder and Telemark / Southern parts of Agder | | Southern parts of Rogaland, Agder and Telemark | | | |
| 29th | Southern parts of Rogaland, Agder and Telemark | Southern parts of Rogaland, Agder and Telemark | | | | | |
| 28th | Agder and Telemark | | | | | | |
| | **30th** | **1st** | **2nd** | **3rd** | **4th** | **30th** | **1st** |
| | | | **Day for which the warning was valid for** | | | | |

The landslides thresholds for the area showed high awareness level in Agder, for a period of two-three days. The rail and road authorities were warned already on Thursday 28th in an e-mail, before the first warning was issued. The rail and road authorities used this early information to plan the possible closure of railways and to plan the use of extra personnel.

The first issued warning, on Friday 29th of September, was a yellow level (the lowest warning level) for the days of Saturday 30th of September and Sunday 1st of October (Table 3). On Saturday 30th September, the level for landslide hazard was upgraded to orange level and kept orange for the two following days for parts of the Agder counties, while the rest of the area including also the county of Telemark had a yellow level (Fig. 5; Table 3). The hazard for Telemark was levelled down to green on Monday 2th of October, but Rogaland was on a yellow level for one day longer than the Agder counties, until the 3rd of October. Based on these warnings the regional offices of NVE were activated and started to communicate and inform the respective counties and municipalities to check on the implementation of the emergency plans.

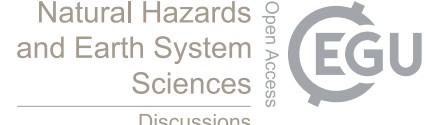

**Figure 5.** Issued landslide warnings for southern Norway in the period from 30/9 to 3/10, 2017 (source: xgeo.no).

During these rainfall events that lasted from 29[th] of September to 3[rd] of October some of the rain gauges in the area (the Agder counties) received nearly 300 mm in 3-4 days. Many rain gauges measured precipitation that corresponds to more than 100

5    years return period rain (Gislefoss et.al, 2017).

The first intense rainfall started around midnight the 29[th] of September along the coastline of Rogaland, moving eastward in Agder counties the following hours. Most of the rainfall felt the 30[th] of September and until the 1[st] of October until 4:00h in the morning. After a break during the day of 1[st] October, the second strong low pressure system arrived in the evening of the 1[st] of October around 22:00h and the most intense rainfall felt until very early in the morning the 2[nd] of Ocober (7:00-8:00h).

10    These rainfall events triggered extensive floods and many landslides (Fig. 6) mainly in the counties of Vest-Agder and Aust-Agder, but some landslides occurred also in Rogaland and Telemark. The 80 % of the damages were on private buildings and many people had to evacuate. More than 3300 cases of damages were reported for a value of 500 mill NOK (~50 mill €) (Holmqvist and Langsholt, 2017).





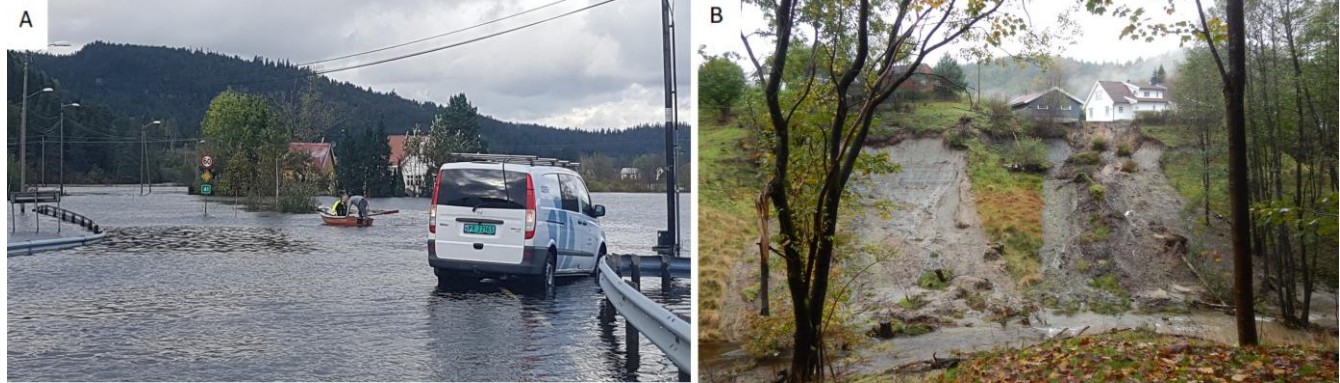

**Figure 6.** Examples of floods and landslide in Southern Norway, September-October 2017. a) Flooded county road by river Tovdalselv, at Drangsholt, Kristiansand. 01.10.2017 (Photo: Turid Haugen, NVE). b) Shallow debris slide, close to house in Augland Kristiandsand.

03.10.2017 (Photo: Ellen Davis Haugen, NVE).

The observed return period of flood was between 50-100 years in many of the large rivers of Agder. However, in some others rivers, the flood was even rarer with longer return periods. In the Mandal River, for instance, the flood was the highest registered since 1896. For many of the stations in this region, with long time series, this was the largest flood ever recorded

(Holmqvist and Langsholt, 2017) (Fig. 6a; Fig. 7).

A still preliminary registration (the verification is still in progress), shows that around 60 landslides events occurred between 29[th] of September and 2[nd] of October in the counties of Rogaland, Agder and Telemark (Fig. 8). They were reported along the main roads, causing blockage, but also houses were directly affected. The landslides registered were mainly shallow soil slides, planar slides, but also rotational and planar slides in clay materials, mainly of marine origin (Fig. 6b). NVEs regional engineers

were attending several of the landslide sites, especially those close to houses, and gave advices to the affected residents, the Police and the municipalities. Since of the presence of marine clay deposits in this area, one of the main concern was the fear that some of the small soil slides could develop catastrophically in quick clay slides. Most of landslides occurred during the most intense rainfall, especially during the 30[th] of October and in the night between 1[st] and 2[nd] of October.





**Figure 7.** Issued flood warnings for southern Norway in the period from 30/9 to 3/10, 2017 and water discharge observations. The maps shows also places where roads were closed because of flood (triangles). Discharge stations are also shown, where the discharge level is classified based on how large the flood was (source: xgeo.no).

The newly updated landslide thresholds for southernmost part of Norway, included Agder counties, were very useful in this situation. This made the forecasters more confident that the high awareness level was necessary. However, because the recent correction of the thresholds we did not have long experience with the new thresholds, making the hazard assessment more complicate (Fig. 8). A daily communication between the regional engineers from NVE and the landslide forecaster on duty
10  help to understand the ground conditions at local scale.

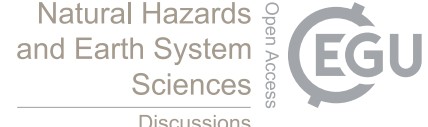



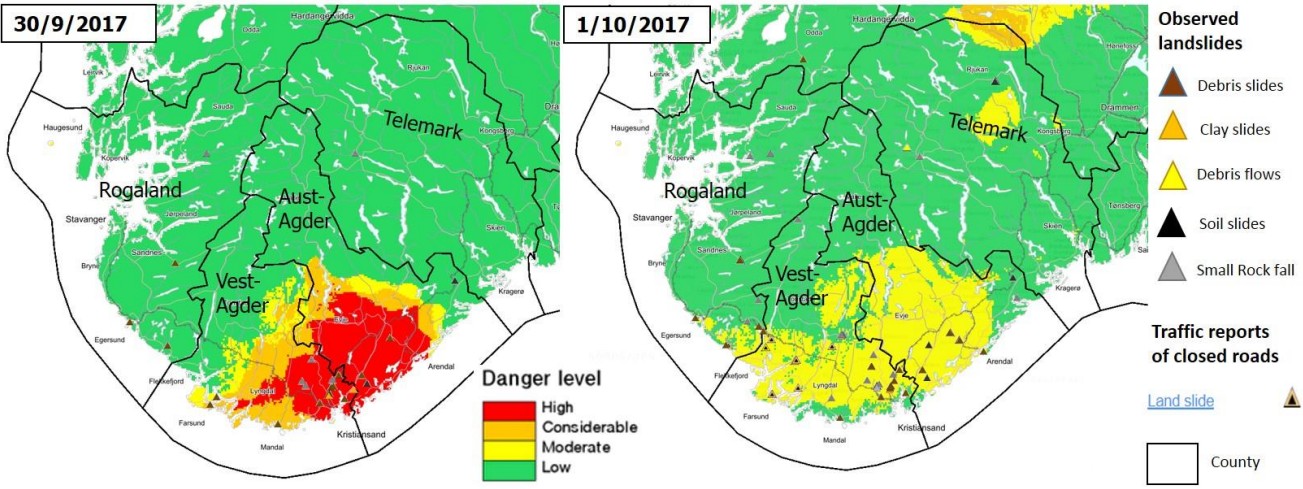

**Figure 8.** Observed landslide threshold maps (Hydmet Sørlandet) and distribution of landslide events. Closed roads due to landslides are also viewed in the same map (source: xgeo.no).

Thanks to the issued landslide and flood warnings the public and municipalities became more aware of the severity of the situation, before and during the event. Therefore they were more prepare to face damages, and the closure of roads and railways.

## 8 Conclusions

The development of the rainfall- and snowmelt-induced landslide forecasting and warning service in Norway was possible thanks to a joint initiative across governmental agencies, and thanks to the fact that we could take advantage of existing IT-
tools, hydrological models and hydrogeological network available at NVE as part of the well-established flood forecasting service.

The joint initiative with the MET, NPRA and Bane NOR was crucial for the establishment of the service and it is still important for the operation of the service (in terms of economy, collection of landslide events, common research and development). The synergy with the flood warning service was significant for a rapid establishment and a rational operation (organization,
hydrological monitoring and models, automatic collection of MET observations & forecasts, decision tools, warning routines & communication).

The landslide forecasting and warning service is herein presented, describing main components, warning levels, routines for daily assessments, web tools for communication, etc. The service uses real-time measurements of hydro-meteorological data (i.e. discharge, groundwater level, soil water content and soil temperature, snow water equivalent, meteorological data), and
model simulations of the meteorological and hydrological conditions. The thresholds used, are based on statistical analyses of historical landslides and simulated hydro-meteorological variables (such as rainfall, snowmelt, soil saturation and depth of frozen ground) and shown as a hydro-meteorological index. The service identify potentially dangerous situations, and notifies



local emergency authorities and the public up to 66 hours ahead with the purpose that they can take preventive measures. A case study from autumn 2017 has been presented showing how the service is well functioning and useful in order to prevent and reduce damages due to landslides and spare lives. NVE is a public administrator, but at the same time, it is the institution nationally responsible for water resources in Norway. Therefore, a lot of research and technological development has been

5    carried out in the past by the Hydrology department at NVE. This tradition of combining research, management and administrative tasks has been an advantage when developing the landslide forecasting service. Most of the research and tools have been developed by staff at the NVE, however some research and IT-development has been out-sourced, but mainly performed under supervision from NVE staff. Anyhow, NVE acknowledge the benefits by having forecasters, researches and developers closely connected.

10   The first results after four years of operations indicate that the flood and landslide services have succeeded as a tool for the road and railway authorities in increasing awareness, preparedness and risk reduction. NVE's user survey from 2016/2017 (Epinion AS, 2017), confirms that warnings issued by NVE (flood, landslide, snow avalanche) is considered as an "alarm clock" for the municipalities, contingency planners. The service is wanted and appreciated by our most important users, mainly the responsible of emergency response at municipalities and counties and the police.



Our aim and strategy is to provide correct forecasts of both spatial and temporal landslide occurrence and systematically updated landslide bulletins. Therefore, we need:

- Reliable weather forecasts

- Reliable real-time data and hydrological models

- Long-term records (data/events) and good hydrological statistics.

- Good quality landslide data

- Roles and responsibilities well defined, and agreed cooperation with key agencies

- Good internal and external coordination

- Precise and understandable communication

- Continuous evaluation, research and development, improvement

- Skilled and experienced personnel

Even if the service is quite satisfactory, hazard assessment, tools for decision-making (xGeo, hydrological models, indexes and thresholds) and communication (varsom.no) need to be continuously developed and improved. Many are still the

challenges and limitations, and we work continuously for future improvements. To improve the accuracy, precision and usefulness of the service, the following areas should be strengthened (Colleuille et al. 2017):

**Hazard assessment:** The usefulness of the issued warning can be increased considerably if combining landslide hazard and vulnerability data. Therefore hazard and risk maps, represent important tools for local authorities assisting them to set priorities and where to implement the required measures. However hazard maps and risk maps are not available in the Norwegian

municipalities, therefore landslide susceptibility maps available for the entire country could be used by local emergency authorities. These maps have been used to improve the thresholds, but they could support the municipalities showing where landslides may occur. We need to communicate better to the users the importance of such maps, in lacking of other hazards maps.

**Weather forecasts and hydrological models:** Reliable warnings require reliable meteorological observations and forecasts.

Landslide triggered by summer rain showers still represent a challenge to be predicted. The cooperation between MET and NVE has contributed to improved grid data (precipitation and temperature) of observations and forecasts, thus improving estimates of snow, water flow and other hydro-meteorological variables.

The hydrological model used to calculate the water saturation, a parameter used in landslide thresholds, still has a rough resolution in both time (24 hours) and in space. The model uses input grid data of precipitation and temperature (observations

and forecasts) based on a rough interpolation and does not utilize yet the improved grid data provided by MET at the end of 2016 (Lussana et al. 2017; Saloranta 2016). An improved version of the model is scheduled to operative in 2018/2019. It is



also appropriate to implement three-hour resolution. Therefore, there is still considerable potential for improving the basis used for landslide thresholds.

**Better verification of landslide events**: The service requires of reliable landslide events (e.g. correct type, date, place, triggering). This is a prerequisite both for establishment of thresholds and post-evaluation. For the first evaluation, it is enough to know if landslides have occurred, but to tune the warning levels it is important to know how many landslide occurred under a specific warning level. It is essential to have a good overview of the number and dimension of occurred landslide events after a rainfall and/or snowmelt episode. NVE maintain a national database, however, the registration is still sparse, i.e. there is no systematic record of events in all regions of Norway. The quality of registrations also varies greatly. There is not consistency in data collection and there are problems with the classification of different landslide types. Release time and location can also be wrong and triggering causes are not reported.

A major challenge for the verification of the landslide occurrence, is that we rely on media, and not on systematic field observations. Also we do not have systematic routines to collect data especially for those events on buildings or outside main road and railways. Quite often small events are not reported. Systematic aerial photos campaigns or the use of satellite images after rainfall events could be very useful to improve landslide mapping. Works are in progress on this topic involving master students from different Norwegian universities.

Upgrading of landslide inventories is mandatory after each forecast, in order to have the correct type but also number and dimensions of landslide events. Results from a recent international workshop (Devoli, 2017) indicated that it is important also to let stakeholders know about the importance of landslide registration and maintenance of inventories, what data are necessary to collect, and make them familiar with the need to ensure reliable data, e.g. what type of landslide.

In Norway most of the data along roads (which is the majority of events) are not recorded by specialists, and hence there is a degree of uncertainty in the quality of the data. The NPRA is now working to ensure that all contractors responsible for driving roads user standard format and receive periodic training. Because of the poor quality of landslides data and lack of observations, it has been a challenge to tune the landslide thresholds in some regions.

**Increase communication and build the user's capacity:** The greatest opportunity to increase the benefit of the service is to build expertise among users. Some of the challenges are: communicating the warning on time and with sufficient leading time to take actions and the communication of the uncertainty. We have observed that rainfall- and snowmelt-induced landslides are often considered as flood damages (i.e. debris flows and debris slides/avalanches) or snow avalanche damages (i.e. slushflows). There is a need to strengthen the dissemination work specifically aimed at regional and local authorities, but also the public and the media, so that the warning service itself, the background for alerts and the different landslide types are better understood. The goal is also to get users and recipients of warnings to contribute significantly more with for example registration of hazard signs and landslide events.





## Acknowledgments

A special thanks to our partners: The Norwegian Meteorological Institute (MET), the Norwegian Public Road Administration (NPRA) and the Norwegian Rail Administration (Bane NOR).

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
