# Peer review of "The Norwegian forecasting and warning service for rainfall- and snowmelt-induced landslides"

_Natural Hazards and Earth System Sciences, 2017_

## Referee Comment (RC1) · Anonymous Referee #1 · 15 Jan 2018

The paper "The Norwegian forecasting and warning service for rainfall and snowmelt-induced landslides" deals with the geo-hydrological EWS in use in Norway and encompasses a detail description of its functioning. The organization of the paper is good and the manuscript itself is well-structured with meaningful images and tables. The paper is quite long and therefore I suggest at least to remove section 3.1, which is interesting but not necessarily relevant for the NHESS audience. If so, you can write the joint composition of the service in the introduction, since you mention it in the abstract.

On the other hand, the authors touch some very interesting topics but do not delve into them. I recommend furnishing some explanations concerning the following important

points: 1) First, I recommend adding a chart describing the "communication chain", ie to show all the passages from the moment data are acquired and a forecast is made to the final recipients (the population). Also, who is responsible for the communication to the citizen? The mayor? 2) How do you reach the population? Only by voluntary subscription to SMS and email? Is there a TV or radio broadcast? Sirens or cars passing by and giving the alarm? Automatic SMSs to the people in a certain area (even to people who did not subscribed to the notification system and even tourists that do not live in the area)? Smartphone apps? If none of these methods are used, how can you reach a significant percentage of the population with your warnings? 3) I recommend adding an example (probably as a new figure) of a bulletin that you send to the population and/or to the local administrations. Is there an intermediation of the local administrations? If so, when you communicate an alert, do you use the same language for both administrations and population or is the communication to the administration more technical and to the population simpler? 4) It seems that your yellow alert is not very conservative. Some Countries have a similar system, but yellow alerts are issued as many as 100-150 times every year, thus creating obvious false alarms issues. Do you have false alarms problems or maybe the opposite (missed alarms)? How do you cope with false or missed alarms?

The language is generally good although there are ubiquitous errors especially concerning singular and plural forms. I have corrected the text when I spotted them but, since I probably missed some of them, all the authors should carefully re-read all the paper paying particular attention to this issue.

These and other recommendations are listed below:

Table 1: in the warning box change "allow" into "allows". Page 3, line 4: replace the semicolon with a comma. Page 3 line 24: replace hazards with hazard and years with year. P3 l26: replace dike with dikes. P3 l28: add "it" after "that". P3 l29: replace options with option. P4 l20: replace lies with lie. Also add "are" before "to be found". P4 l28: replace fall with falls and freezes with freeze. P4 l29: add "the" before "decrease"

and "increase". Also, add the final "s" to "contribute" and "river". P4 l30: replace rises with rise. P4 l32: replace includes with include. P5 l2: remove "s" from "frequents". P5 l7: "gives" P5 l7: replace "the north of" with "Northern" for similarity to "Western" used before. P5 l9: please explain ice jams in the text. P5 l18: "loose" not "loos". P6 l6: lives P6 l21: replace mill with million and add in brackets the equivalent in USD. P6 l22: numbers P6 l 32: operates Section 3.1 is interesting but not necessary relevant for the NHESS audience. Consider removing it. If so, you can write the joint composition of the service in the introduction, since you mention it in the abstract. P7 l27: needs P7 l28: define 8, 2: uses 8, 2-6: the authors should provide references fort these models and/or furnish a brief explanation. 8, 19: runs 8, 23: add "the" before "parametrisation". 9, 14: when does the inventory date back to? 9, 30: today's instead of todays. 9, 28 – 10, 2: this part is not clear. Please explain better. 10, 24: landslides 11, 15: delete "in". 12, 7: flood forecasters 12, 8: leads 12, 13: if you state that forecasters are always available one would think of a 24h availability. So remove "always". 12, 15: "Forecasters, when on duty" 12, 31: "twice a day" 13, 5: are available 13, 11: you talk here about orange and red levels, but the reader still does not know what they are. Also, here you explain who forwards the message to whom. I recommend adding a table showing the complete chain of communication from those who provide data and forecasts to the citizen. 13, 14: please furnish an explanation of what CAP is. 15, 4: consists Table 2: change "infrastructure" into "infrastructures" in the red level box. 17, 8: implies 17, 16: relies 17: 18: warns 17, 20-22: this sentence is not clear. Please rephrase. 18, 1: the first "How" should not be written in lower case. 18, 2-4: here I suggest inserting a contingency table with the number of events predicted and occurred (true positive), predicted but that eventually did not occur (false negative), unpredicted and not occurred (true negative) and unpredicted but occurred (false positive). 18, 22. Add a full-stop before "Results" and change "shows" with "show" and "consider" with "considers" (in the latter case the subject is "majority", which is singular). 18, 26: replace "conducted"" with "carry out". 19, 2: includes 24, 22: identifies 26, 17: replace "if" with "by" 27, 11: remove both commas from this line.

---

## Referee Comment (RC2) · Anonymous Referee #2 · 27 Jan 2018

The paper "The Norwegian forecasting and warning service for rainfall- and snowmelt-induced landslides" gave detailed descriptions of the history, developments, scientific results, applications and case study of the warning service. Besides, this paper not only addressed essential components of Early Warning System (EWS) e.g. meteorological forecasts, hydrological models, landslide database, thresholds and verifications, but also pointed out some challenges that need to be overcome. Therefore, this paper fitted in with this special issue as well as NHESS journal.

However, there were still several issues need to be clarified/revised before publication:

(1) What types of landslide does this EWS deal with? Considering different types of

landslide (e.g. deep-seated landslide, shallow landslide, rockfall, debris flow or any others) should refer to different early warning models, if there were not just only one type of landslide included in this EWS, the authors might need to add a table to tell readers the types of landslide, the early warning models and the parameters used in this system.

(2) Page 6, Line 3∼4: The authors listed several dates of important landslide events, however, readers might not be able to understand how important or how special they were. A table with more detailed information such as rainfall intensity, accumulated rainfall, duration, geological and geomorphological conditions might need to be added.

(3) Section 3: Although several references were provided throughout section 3, the authors are recommended to briefly explain some important methods such as warning models and thresholds with equations or figures so that the readers might be able to get a picture of the theory running in the system more quickly.

(4) Page 9, Section 3.2.4: It is suggested that at least probability of detection and probability of false alarm should be demonstrated here or in section 5.

(5) Page 18, Line 1∼4: The authors mentioned that over 95% of the assessment were considered as correct during 2013-2016, but the methods and data for the validations of threshold values, hydrological simulations, weather forecasts and the judgements of forecasters in duty were not shown here.

(6) Page 20, Table 3: How many days ahead does this EWS provide landslide/flood forecasting warnings? In section 3.2.1, the authors mentioned that AROME forecasts the weather for the next 66 hours while EC forecasts for nine days ahead. On the other hand, in page 11, line 9∼10, the authors said forecasting weather are provided for the next 6 days on the website Xgeo.no while in page 12, line 31, the authors said forecasting warnings are provided for the next 3 days on the website varsom.no. If this EWS provides forecasting warnings only for the next 3 days, then it is suggested that the days beyond 3 days should not fill in green color in table 3 since this might lead

to the misunderstanding that warning level of 4th Oct. can be predicted on 28th Sep. Besides, in the table of flood warning, the "day for which the warning was valid for" might be typos because 4th followed by 30th and 1st.

---

## Author Comment (AC1) · 9 Feb 2018

We inform that we have read the feedbacks and comments provided by the two reviewers and we would like to thank them for their careful review and their valuable comments. We have appreciated all the comments and suggestions provided. We agree with the comments. We consider them very constructive and useful to improve the quality of the manuscript. We will try to address all suggestions preparing a new version of the manuscript. Our replies to general and specific comments of Reviewers #1.

Anonymous Referee #1

[Figure]

General comment / remark:

The paper "The Norwegian forecasting and warning service for rainfall and snowmeltin-duced landslides" deals with the geo-hydrological EWS in use in Norway and encompasses a detail description of its functioning. The organization of the paper is good and the manuscript itself is well-structured with meaningful images and tables. The paper is quite long and therefore I suggest at least to remove section 3.1, which is interesting but not necessarily relevant for the NHESS audience. If so, you can write the joint composition of the service in the introduction, since you mention it in the abstract.

Respond: We thank Referee #1 for these feedbacks and comments. We agree that the manuscript is quite long, and we see now that section 3.1 is not very relevant for most of NHESS's readers. We will try to include just the essential information from section 3.1 in the introduction, and then remove the whole section 3.1.

On the other hand, the authors touch some very interesting topics but do not delve into them. I recommend furnishing some explanations concerning the following important points:

The communication chain 1) First, I recommend adding a chart describing the "communication chain", ie to show all the passages from the moment data are acquired and a forecast is made to the final recipients (the population). Also, who is responsible for the communication to the citizen? The mayor?

R: Thank you for this comment. We see now that this topic is not enlightened well enough. In Norway, the County have the responsibility to forward the warnings to the mayors and/or contingency responsible of the different municipalities, especially for the two highest levels, orange and red. However, in 2017, we gave everyone the possibility to subscribe and now the mayor also has the possibility to receive directly warning messages, therefore the communication chain has changed. A chart that describes the chain of communication of the warning bulletins will be very useful for the readers. We will include it in the manuscript. We will also explain who is responsible to inform

the citizens that a landslide warning is issued for their municipality.

How do you reach the population? 2) How do you reach the population? Only by voluntary subscription to SMS and email? Is there a TV or radio broadcast? Sirens or cars passing by and giving the alarm? Automatic SMSs to the people in a certain area (even to people who did not subscribed to the notification system and even tourists that do not live in the area)? Smartphone apps? If none of these methods are used, how can you reach a significant percentage of the population with your warnings?

R: Thank you for this comment. We understand that the subject concerning communication to the public/citizens is interesting for the readers. We use several communication platforms to reach the population. We will give some greater knowledge of this in the manuscript.

An example of a bulletin 3) I recommend adding an example (probably as a new figure) of a bulletin that you send to the population and/or to the local administrations. Is there an intermediation of the local administrations? If so, when you communicate an alert, do you use the same language for both administrations and population or is the communication to the administration more technical and to the population simpler?

R: Our bulletins are published at the web portal www.varsom.no, and the local administrations and others that subscribe to the notification system gets an e-mail or SMS with a short message that let them know it is a warning issued and with an URL directly to the relevant warning bulletin on www.varsom.no. We only write one version of the bulletin, in the same language. We strive to keep the language simple enough for the public, but at the same time sufficient for the local administrations/alarm personnel. Since the warnings are regional, there is not very detailed technical information provided in the bulletin. We recommend the public and the emergency authorities to make local assessment and decide the measures most appropriate, but the warning service is available for consulting by phone or e-mail. The bulletins are in Norwegian, but an English version was launched in January 2018. We will add an example of a bulletin in

the manuscript. Thank you for this comment.

False and missed alarms 4) It seems that your yellow alert is not very conservative. Some Countries have a similar system, but yellow alerts are issued as many as 100-150 times every year, thus creating obvious false alarms issues. Do you have false alarms problems or maybe the opposite (missed alarms)? How do you cope with false or missed alarms?

R: False or missed alarms is of course an issue also for the Norwegian early warning system. Our system is an expert knowledge centred-system. The expert analysed daily the thresholds and decide which warning level to send. We experienced short time after we were operational, that the threshold values for the yellow alert in some areas were too sensitive. This occurred especially in some regions or in summer with short and intense rain due to their spatial and quantitative uncertainty. In addition, because of our lack of experience, we relied too much on our thresholds. The result were too many false alarms. In the last two years we have adjusted the threshold values for two regions in Norway (The South of Norway; the area of case study in the manuscript, and the Eastern of Norway). The new and improved thresholds, together with the acquired warning experience, allow us to reduce the number of false alarms. In addition, we have lately improved the cooperation with the Met Office in Norway, working together in the preparation of prediction tools for flash floods and landslides due to heavy and intense rainfall (like thunderstorms) in summer. The Met Office issue a warning on heavy and intense rain and local flash flood and/or landslide hazard in cooperation with NVE. The results of this collaboration were concretely observed last summer when the number of false alarms was notably reduced.

Thank you for the useful remark. We will add some information on missed and false alarms and the accuracy of the forecasting service in the manuscript. Referee #2 has also commented on this.

Language The language is generally good although there are ubiquitous errors especially concerning singular and plural forms. I have corrected the text when I spotted them but, since I probably missed some of them, all the authors should carefully re-read all the paper paying particular attention to this issue. These and other recommendations are listed below:

Table 1: in the warning box change "allow" into "allows". Page 3, line 4: replace the semicolon with a comma. Page 3 line 24: replace hazards with hazard and years with year. P3 l26: replace dike with dikes. P3 l28: add "it" after "that". P3 l29: replace options with option. P4 l20: replace lies with lie. Also add "are" before "to be found". P4 l28: replace fall with falls and freezes with freeze. P4 l29: add "the" before "decrease" and "increase". Also, add the final "s" to "contribute" and "river". P4 l30: replace rises with rise. P4 l32: replace includes with include. P5 l2: remove "s" from "frequents". P5 l7: "gives" P5 l7: replace "the north of" with "Northern" for similarity to "Western" used before. P5 l9: please explain ice jams in the text. P5 l18: "loose" not "loos". P6 l6: lives P6 l21: replace mill with million and add in brackets the equivalent in USD. P6 l22: numbers P6 l 32: operates Section 3.1 is interesting but not necessary relevant for the NHESS audience. Consider removing it. If so, you can write the joint composition of the service in the introduction, since you mention it in the abstract. P7 l27: needs P7 l28: define 8, 2: uses 8, 2-6: the authors should provide references fort these models and/or furnish a brief explanation. 8, 19: runs 8, 23: add "the" before "parametrisation". 9, 14: when does the inventory date back to? 9, 30: today's instead of todays. 9, 28 – 10, 2: this part is not clear. Please explain better. 10, 24: landslides 11, 15: delete "in". 12, 7: flood forecasters 12, 8: leads 12, 13: if you state that forecasters are always available one would think of a 24h availability. So remove "always". 12, 15: "Forecasters, when on duty" 12, 31: "twice a day" 13, 5: are available 13, 11: you talk here about orange and red levels, but the reader still does not know what they are. Also, here you explain who forwards the message to whom. I recommend adding a table showing the complete chain of communication from those who provide data and forecasts to the citizen. 13, 14: please furnish an explanation of what CAP is. 15, 4: consists Table 2: change "infrastructure" into "infrastructures" in the red level box. 17,

8: implies 17, 16: relies 17: 18: warns 17, 20-22: this sentence is not clear. Please rephrase. 18, 1: the first "How" should not be written in lower case. 18, 2-4: here I suggest inserting a contingency table with the number of events predicted and occurred (true positive), predicted but that eventually did not occur (false negative), unpredicted and not occurred (true negative) and unpredicted but occurred (false positive). 18, 22. Add a full-stop before "Results" and change "shows" with "show" and "consider" with "considers" (in the latter case the subject is "majority", which is singular). 18, 26: replace "conducted"" with "carry out". 19, 2: includes 24, 22: identifies 26, 17: replace "if" with "by" 27, 11: remove both commas from this line.

R: Thank you very much, Referee #1, for the careful review of the language in the text. We will correct the errors, read carefully through the text again, and improve the language. In addition, we will answer the other questions you raise here, like when to the landslide database date back.

---

## Author Comment (AC2) · 9 Feb 2018

We inform that we have read the feedbacks and comments provided by the two reviewers and we would like to thank them for their careful review and their valuable comments. We have appreciated all the comments and suggestions provided.

We agree with the comments. We consider them very constructive and useful to improve the quality of the manuscript. We will try to address all suggestions preparing a new version of the manuscript.

Our replies to general and specific comments of Reviewer #2.

[Figure]

Anonymous Referee #2

Overall comment The paper "The Norwegian forecasting and warning service for rainfall- and snowmeltinduced landslides" gave detailed descriptions of the history, developments, scientific results, applications and case study of the warning service. Besides, this paper not only addressed essential components of Early Warning System (EWS) e.g. meteorological forecasts, hydrological models, landslide database, thresholds and verifications, but also pointed out some challenges that need to be overcome. Therefore, this paper fitted in with this special issue as well as NHESS journal.

We thank Referee #2 for this positive comment. We are glad to hear that our work is interesting for colleagues outside Norway and the NHESS readers.

However, there were still several issues need to be clarified/revised before publication:

(1) What types of landslide does this EWS deal with? Considering different types of landslide (e.g. deep-seated landslide, shallow landslide, rockfall, debris flow or any others) should refer to different early warning models, if there were not just only one type of landslide included in this EWS, the authors might need to add a table to tell readers the types of landslide, the early warning models and the parameters used in this system.

Respond: Thank you for this important comment. In Norway, it yearly occurs a large variety of landslides, but the Norwegian landslide EWS, described in the manuscript is designed to warn only rainfall- and snowmelt induced landslides. Under this general name we include, soil slides (e.g. clay/silt slides), debris slides (e.g. gravel/sand/debris slides), debris avalanches, debris flows, debris floods as defined by Hungr et al. (2014). In addition to these, we warn also slushflows, described in Hestnes (1985). We use the same thresholds for the all of them. However, we know that soil saturation is significantly for soil slides, but the amount of rain and snowmelt is more important for the triggering of debris flows. We will include a figure that illustrates this in the manuscript. However, for slushflows we have defined other separate thresholds in based on both

weather, snow (depth and structure) and hydrological conditions (as frost). We are still working on improving the thresholds for slusflows. Rock falls, clay slides, quick clay slides and rock avalanches are not included in this EWS. We will address this issue and try to explain it better in the manuscript.

Major floods and landslides in Norway

(2) Page 6, Line 3-4: The authors listed several dates of important landslide events, however, readers might not be able to understand how important or how special they were. A table with more detailed information such as rainfall intensity, accumulated rainfall, duration, geological and geomorphological conditions might need to be added.

R: Thank you for this remark. We agree. We will work on a better and more interesting presentation of the historical events.

The Norwegian landslide forecasting and warning service

(3) Section 3: Although several references were provided throughout section 3, the authors are recommended to briefly explain some important methods such as warning models and thresholds with equations or figures so that the readers might be able to get a picture of the theory running in the system more quickly.

R: Thank you. This part need a better explanation. We can present the figure with our thresholds or eventually other diagram explaining the methods used.

Thresholds

(4) Page 9, Section 3.2.4: It is suggested that at least probability of detection and probability of false alarm should be demonstrated here or in section 5.

R: The chapter you are considering are "3.2.4 Thresholds" and "5 Validation of the forecasting service". We understand that this is an interesting topic for the readers and that we have not explained it enough. We will work on a way to address this in the manuscript without making it too long. Thank you for this comment.

Technical performance

(5) Page 18, Line 1-4: The authors mentioned that over 95% of the assessment were considered as correct during 2013-2016, but the methods and data for the validations of threshold values, hydrological simulations, weather forecasts and the judgements of forecasters in duty were not shown here.

R: Thank you for this comment. We see that this topic ought to be discussed in the manuscript. We will work on a section where we explain how the daily evaluation of performance is made.

The case study

(6) Page 20, Table 3: How many days ahead does this EWS provide landslide/flood forecasting warnings? In section 3.2.1, the authors mentioned that AROME forecasts the weather for the next 66 hours while EC forecasts for nine days ahead. On the other hand, in page 11, line 9-10, the authors said forecasting weather are provided for the next 6 days on the website Xgeo.no while in page 12, line 31, the authors said forecasting warnings are provided for the next 3 days on the website varsom.no. If this EWS provides forecasting warnings only for the next 3 days, then it is suggested that the days beyond 3 days should not fill in green color in table 3 since this might lead to the misunderstanding that warning level of 4th Oct. can be predicted on 28th Sep. Besides, in the table of flood warning, the "day for which the warning was valid for" might be typos because 4th followed by 30th and 1st.

R: Thank you for this comment. We see now that this theme was not adequately described and that it can be confusing. We will work on improving this section. When we talk about the "The next six days" we meant 3 days + 6 days (the period of the short-term prognosis, and the further 6 d from the long-term prognosis). Still, we only publish awareness levels for landslide for three days (today, tomorrow and the day after tomorrow), since the short term prognosis is more reliable than the long-term prognosis. We will correct this in the new manuscript and we will try to visualize better

the dynamic of prognosis and warnings. And thank you for pointing out the writing error in table 3.

---

## Referee Comment (RC3) · Anonymous Referee #1 · 12 Feb 2018

Thank you for your answers. I will carefully read the updated paper as soon as you upload it.
* * *

---

## Referee Report (RR1)

I would like to thank the authors took all the comments into consideration and made a detailed revision of the manuscript. However, some issues/sentences are still needed to be revised before the publication.

(1) [Page 12, Figure 2] How to decide the threshold of yellow, orange and red? Is it manual? If yes, what is the rule and how to make it more objective?

(2) [Table 4] The summations of percentage are not 100% in 2013, 2014, 2016, especially, 100.2% in 2016.

(3) [Page 24, Line 7-8] The authors said that "The results of the preliminary analysis conducted at the national scale for the period 2013-2016 and using all days in the years, shows that over 95% of the days assessment are considered as correct." However, the percentage of correct prediction in 2013 and 2014 are 94.2% and 92.9% respectively in Table 5.

(4) [Page 11, Line 26] It is suggested that "Figure 2 a shows the first version of thresholds" might need to revise as "Fig. 2(a) shows the first version of thresholds" for not confusing "(a)" from "a".

(5) [Page 11, Line 26-27] "Figure 2 a shows the first version of thresholds at national level, however, defined using three major landslide events. However, there are regional differences in the prevailing types" It seems to be a little weird when two "however" show successively.

(6) [Caption of Figure 4] "chapter 6" might be "section 6"?

(7) [Page 23, Line 22] "Challenging days" -> "challenging days"

(8) [Page 24, Line 19 and 21] "km2" should be superscript.

---

## Author Response (AR2)

**Replies to comments of Editor and Referee #1 and #2**

Submission ID: nhess-2017-426

April 2018

We would like to thank the editor and reviewers for being positive to reconsider our manuscript after a minor revision with the suggestions from all three in mind. Thank you for the opportunity to send you a final manuscript.

We believe to have done our best to improve the manuscript, and hope that both editor and reviewers agree that it is sufficient to be published in the NHESS special issue on landslide early warning. We thank you for the careful reviews and positive feedbacks.

Our revisions are marked up in the manuscript. We have done some additional correction of spelling and tried to shorten some of the sections. The section on thresholds (3.1.4) is quite altered. We hope that Editor approves the revision.

In the following, we will comment what kind of changes we have done in the manuscript, due to the comments from the editor and the two referees.

**Anonymous Referee #1**

The authors addressed well of my comments. I only have few minor indications left:

Page 11, line 33: replace "without observations of landslide" with "without observations of any landslide". Response: Corrected.
16, 4: replace "on" with "a". Correct the double "in". R: Corrected.
22, 19: replace larger with large. R: Corrected.
23, 2: in the brackets where you write "(should be yellow)", you don't really explain what you mean. I think that you mean the case when a warning was actually necessary but the level of severity attributed is wrong.
R: We erased the brackets with "should be yellow" and "should be green", I believe that the sentence is more understandable. Thank you for pointing this out.
 23, 9: I am not sure about the use of the word "assured" here.
R: We changed the sentence a bit, and hopefully it is easier to understand what we mean now. Thank you for this comment.
 24, 14: remove the article "a". R: Corrected
Figure 2, caption: replace "use" with "used". R: Corrected

**Anonymous Referee #2**

I would like to thank the authors took all the comments into consideration and made a detailed revision of the manuscript. However, some issues/sentences are still needed to be revised before the publication.

(1) [Page 12, Figure 2] How to decide the threshold of yellow, orange and red? Is it manual? If yes, what is the rule and how to make it more objective? Response: Thank you for these questions. We have made a major revision of the section 3.1.4 Thresholds, and hopefully this question is now properly answered.

(2) [Table 4] The summations of percentage are not 100% in 2013, 2014, 2016, especially, 100.2% in 2016. R: We apology this error. The table now show correct numbers. Thank you for making us aware of this.

(3) [Page 24, Line 7-8] The authors said that "The results of the preliminary analysis conducted at the national scale for the period 2013-2016 and using all days in the years, shows that over 95% of the days assessment are considered as correct." However, the percentage of correct prediction in 2013 and 2014 are 94.2% and 92.9% respectively in Table 5. R: We see that this sentence was not easy to understand. It was actually not necessary either, since the first sentence after Table 5 covers the same theme. Thank you for this comment.

(4) [Page 11, Line 26] It is suggested that "Figure 2 a shows the first version of thresholds" might need to revise as "Fig. 2(a) shows the first version of thresholds" for not confusing "(a)" from "a". R: Thank you. The references of the figures should now be correct.

(5) [Page 11, Line 26-27] "Figure 2 a shows the first version of thresholds at national level, however, defined using three major landslide events. However, there are regional differences in the prevailing types" It seems to be a little weird when two "however" show successively. R: We agree. One however is now erased.

(6) [Caption of Figure 4] "chapter 6" might be "section 6"? R: Corrected.

(7) [Page 23, Line 22] "Challenging days" -> "challenging days" R: Corrected.

(8) [Page 24, Line 19 and 21] "km2" should be superscript. R: Corrected.

[revised manuscript text omitted]
. In Norway, these types of landslides apparently appears alongside one another during bad weather situations. Figure 2 a shows the first version of thresholds at national level, however, defined using three major landslide events. However, there are regional differences in the prevailing types of landslides because of the different geomorphological and geological conditions, as well as the hydrometeorological triggering conditions (Table 2). Figure 2a illustrates these differences. The plotted landslides from Western Norway are displayed almost separately from landslides from Eastern Norway, and with Telemark and Oppland events somewhat in between. In Table 2, the differences between these regions are also illustrated. In later studies thresholds are adapted to take into account these regional differences (Boje et al., 2017), as is shown in Fig. 2 b c. The approach has been to manually adjust the threshold upwards, based on case studies of weather events that lead to higher landslide hazard in the original thresholds without observations of landslide, thereby causing false alarms. This has been done for the regions Southern Norway and Eastern Norway (Figure 2b c). A problem for these two regions were too few recorded landslides to carry out a statistical viable regression analysis. Instead, based on cases with false alarms, the thresholds were simply increased until no impact was shown in the index map. All thresholds are visualized in form of raster data (with 1 km² resolution) and available at xgeo.no (see chapter 3.1.6). Although, a study of thresholds for different landslide types has not been conducted yet, we consider that this could be of interest to test in the future.

[Figure]

[Figure]

**Figure 2.** The landslide hazard threshold use by the Norwegian landslide early warning and forecasting service. a) National thresholds, b) Regional threshold for Southern Norway and c) Regional threshold for Eastern Norway. ~~The landslide hazard threshold use by the Norwegian landslide early warning and forecasting service. a) National thresholds, defined using three main landslide events in three regions in South of Norway: Eastern Norway (2000), Western Norway (2005) and Oppland and Telemark (2008). The plot show days with events and days with no events. 
[revised manuscript text omitted]
 method is based on the number of registered landslides events and hazard signs for regions of about 10-15.000 km². The principle of this classification is based on the defined awareness level (table 3 and figure 7), stating that it is expected more landslides and possibly higher damages for rare hydro-meteorological situations. It was suggested that for a yellow awareness level it should be reported about 1-4 landslides and/or some hazards signs per 10-15.000 km2, about 6-14 for the orange level and over 14 for the red level. Local, small landslides (1-2) caused by local rain showers be accepted for a green level. As pointed out in Piciullo et al. (2017) the landslide density criterion (i.e. the chosen number of landslides pr. awareness level) affects significantly the performance results. Experience after five years shows that a semi-logarithmic scale of number of landslides, with overlap, may be more appropriate to represent defined awareness levels (e.g. 1-10 for yellow, 5-50 for orange, over 40 for red level)~~

[revised manuscript text omitted]

---

## Author Response (AR3)

**Replies to comments of Editor**

Submission ID: nhess-2017-426

27[th] of April 2018

5  We would like to thank the editor for another careful review of our manuscript. We are thankful for the opportunity to be a part of this relevant and interesting special issue.

We agree with the suggested corrections, and the changes are now made in the manuscript. One more spelling error was noticed in P12, L12 and is now corrected ("colors" to "colours"). Our revisions are
10  marked up in the manuscript.

[revised manuscript text omitted]